# Orchestrating chromosome conformation capture analysis with Bioconductor

Jacques Serizay ®[1] ✉, Cyril Matthey-Doret ®[1,2,3], Amaury Bignaud ®[1,2], Lyam Baudry ®[1,2,4] & Romain Koszul ®[1]

Genome-wide chromatin conformation capture assays provide formidable insights into the spatial organization of genomes. However, due to the complexity of the data structure, their integration in multi-omics workflows remains challenging. We present data structures, computational methods and visualization tools available in Bioconductor to investigate Hi-C, micro-C and other 3C-related data, in R. An online book (https://bioconductor.org/books/OHCA/) further provides prospective end users with a number of workflows to process, import, analyze and visualize any type of chromosome conformation capture data.

Chromosome conformation capture methods (3 C, 4 C, Hi-C, micro-C, …) have become a prevalent approach to investigate the interplay between DNA-related metabolic processes and the 3D folding of chromosomes (e.g. gene regulation, chromosome compaction, DNA repair and rearrangements[1–5]). Computational processing of HiC data has also provided powerful and elegant solutions to several genomic limitations, in particular allowing for robust genome scaffolding[6–9]. Furthermore, the application of Hi-C directly to complex microbial communities allows the characterization of whole microorganism genomes, the identification of mobile genetic elements such as viruses and plasmids and their assignment to their respective hosts[10–13], and characterization of prophages activity[14]. International consortia have emerged to orchestrate efforts to characterize chromosome conformation and nuclear organization across cell types, tissue samples and species[8,15].

Genome-wide chromatin conformation capture assays, such as Hi-C, micro-C or DNAse-C[16–18], yield lists of pairs of interacting genomic loci at a base-pair resolution (stored in pairs files, in which each record describes a single measured contact between two genomic loci), which can be further binned to a window of chosen size and stored in symmetric sparse matrix files (where consecutive columns/rows correspond to consecutive genomic bins). Hi-C data specificities thus largely differ from the typical 1D genomic file formats (e.g. '*.bigwig*' or '*.bed*' files). While a single '*.pairs*' file format has been formally defined by the NIH 4D Nucleome Network[19], three file formats have been independently proposed to store binned matrix files, each generated by a specific processing software. Hi-C Pro generates (i) sparse *matrices* as three column (bin$_i$ / bin$_j$ / count$_{ij}$) text files and (ii) *region* files describing genomic coordinates for each bin[20], Juicer produces '*.hic*' files[21] and '*.(m)cool*' files are generated by the *distiller* pipeline[22]. The '*.hic*' and '*.(m)cool*' formats are binary, multi-resolution, highly compressed and indexed files and can rely on companion libraries (respectively *straw* and *cooler*) to perform random access to a subset of the data. Softwares are also developed to manipulate files in these formats (Juicer tools, Juicebox and Juicebox Assembly Toolbox for '*.hic*' files and HiGlass, cooltools and coolpuppy for '*.(m)cool*' files[23–27], FAN-C for '*.hic*' and '*.(m)cool*' files[28]). These computational solutions provide a dedicated shell command line interface (CLI) or a Python API. However, they are not embedded in a larger, genomics-centric ecosystem. Other softwares, such as HiCExplorer, GENOVA, mariner or HiCUP, also provide additional toolkits for Hi-C exploratory data analysis (EDA)[29–31].

The Bioconductor project focuses on the development of R packages to provide classes, methods and functions dedicated to genomic datasets[32–35]. For this reason, Bioconductor has become the reference ecosystem for in-depth genomics investigation (encompassing most genome sequencing methodologies, genome annotations, single cell omics, multi-omics, etc). However, although core methods exist to represent genomic interactions in R (defined by the *GInteractions* class in the *InteractionSet* package[36]), and a few packages exist to perform statistical analyses to Hi-C data (e.g. comparing chromatin interaction frequency between samples with

[1]Institut Pasteur, CNRS UMR3525, Université Paris Cité, Unité Régulation Spatiale des Génomes, Paris, France. [2]Sorbonne Université, Collège Doctoral, Paris, France. [3]Present address: Swiss Data Science Center, École Polytechnique Fédérale de Lausanne, 1015 Lausanne, Switzerland. [4]Present address: Université de Lausanne, Center for Integrative Genomics, Quartier Sorge, 1015 Lausanne, Switzerland. ✉e-mail: jacques.serizay@pasteur.fr

*HiCcompare*[37]), no standard chromosome conformation capture data structure has been defined so far in R. Furthermore, data import methods to parse Hi-C processed files in R are still lacking. Overall, the lack of a unified methodology surrounding Hi-C data has limited their integration in the powerful genomics-centric Bioconductor ecosystem, particularly compared to other omics approaches. To address these limitations, we formally defined a set of classes to represent chromosome conformation capture data with Bioconductor, and developed a set of tools to process, parse, analyze and visualize this type of data in R. Compared to existing solutions, this approach allows the end user to leverage existing, powerful genomics-centric methodology already implemented in Bioconductor. Here, we cover the key aspects of the chromosome conformation capture analysis workflow and describe the packages used at each step as well as interoperability features in R. We also present an online book (https://bioconductor.org/books/OHCA/) introducing the end user to the installation of the required packages, their specific functionalities and several examples of complete chromosome conformation capture analysis workflows.

## Results
### Data representation

The *HiCExperiment* package implements the *ContactFile* class (encompassing the *CoolFile*, *HicFile* and *HicproFile* classes) to connect to a contact matrix stored on disk in one of these three formats (Fig. 1, Fig. S1), supporting Hi-C, micro-C and other 3C-related data binned at a fixed resolution. A *ContactFile* instance also lists the resolutions available in the matrix file and metadata relevant for biological analysis. The *import* method provides random access to a *ContactFile*, to only import relevant chunks of data from large Hi-C matrix files. This instantiates a *HiCExperiment* object containing binned genomic contacts of interest stored as *GInteractions* (Fig. 1). Raw counts and normalized contact frequencies (if available) are automatically imported and stored in a list of *scores*. Additional methods are available to move around the Hi-C map (*refocus*), dynamically change its resolution (*zoom*), subset interactions or add qualitative or quantitative metrics (using the standard '[' subsetting operator and the '$' column accessor), and set/get general information related to the contact matrix (e.g. *seqinfo*, *anchors*, *bins*, *topologicalFeatures*, *metadata*). The *HiCExperiment* package also defines a *PairsFile* class to efficiently import '.pairs' files in R as *GInteractions*.

All the classes implemented in *HiCExperiment* directly extend core Bioconductor classes and generic methods, including *BiocFile*, *GenomeInfoDb* and *GenomicRanges*, ensuring seamless parsing, manipulation and genomic representation of locally stored Hi-C contact matrices in R. Importantly, a *HiCExperiment* object can be seamlessly coerced as a *GInteractions*, a *data.frame* tabular object or a (optionally sparse) matrix. This facilitates its interoperability and the integration of Hi-C processed data with other pre-existing packages.

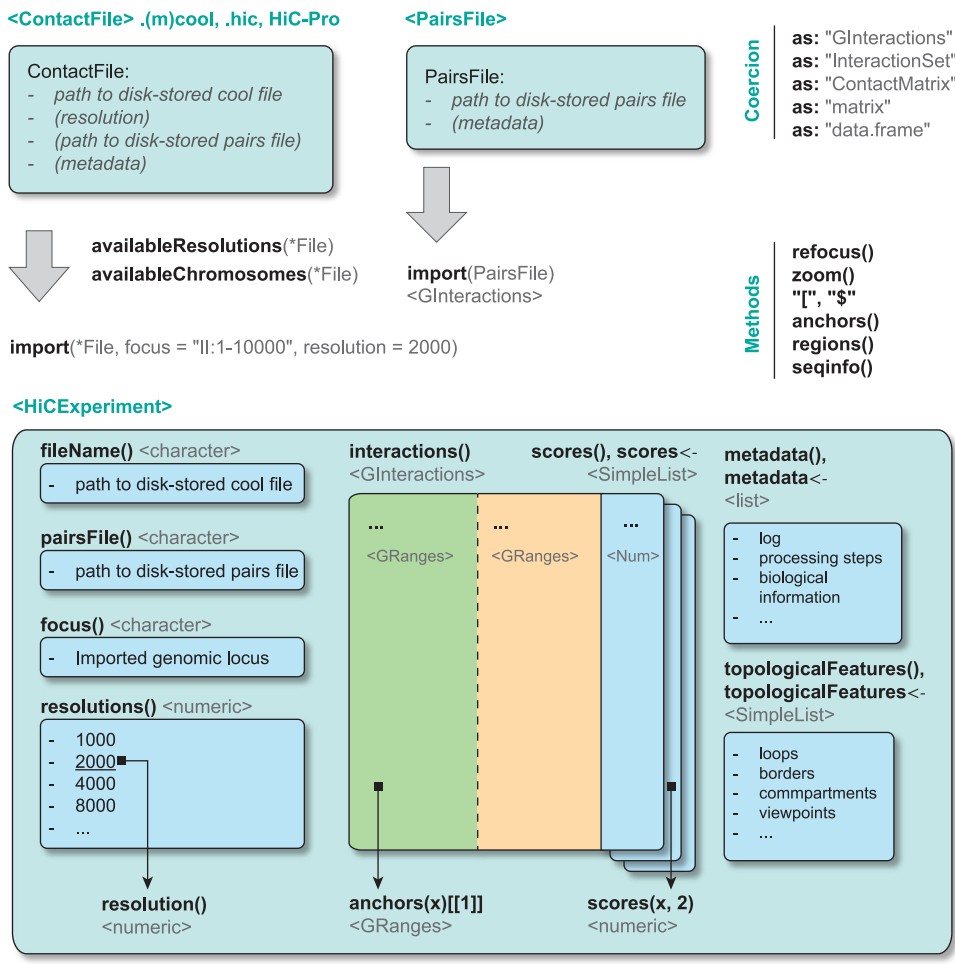

**Fig. 1 | Overview of the ContactFile and HiCExperiment class.** The HiCExperiment package defines the *ContactFile* class, which acts as a pointer to a disk-stored contact matrix in .(m)cool, .hic, or HiC-Pro formats, and the *PairsFile* class, a pointer to a disk-stored tabular pairs file. Both *ContactFile* and *PairsFile* connections can be parsed in R using the generic import function. Importing a *ContactFile* instantiates a *HiCExperiment* object. The main functions and methods provided by the *HiCExperiment* packages to manipulate *ContactFile*, *PairsFile* and *HiCExperiment* objects are written in bold, and the data structure returned by each function/method is indicated between chevron. Interoperability with other R packages is possible thanks to coercing methods provided by the *HiCExperiment* package.

Thanks to ever-decreasing sequencing costs and improving technology, the average size of chromosome conformation capture datasets is continuously increasing, both in sequencing depth and in resolution. HDF5-derived '.(m)cool' and binary '.hic' files both efficiently store such large-scale data, and *HiCExperiment* objects instantiated from these file formats benefit from efficient parsing libraries based on C code, optimized for speed. Furthermore, because random access is supported for these file formats, contact matrices can be partially imported in R, allowing manipulation of large datasets – such as deeply sequenced micro-C datasets – even on personal laptops with standard hardware configuration (e.g. 4 CPUs and 8–16 Gb RAM).

## Data processing

Integrated workflows such as nf-core/hic, open2c/distiller-nf or Juicer[20,21,38,39] efficiently use high-performance computing (HPC) environments to process chromosome conformation capture data. The large number of indirect operations they perform (e.g. container caching, sanitary check-ups and additional quality controls) results in a significant overhead and an increase in storage and memory requirements. These large workflows are therefore less suitable for processing on local workstations, where setting up dependencies is often cumbersome. The *HiCool* package was developed with these limitations in mind. *HiCool* is an R package that automatically sets up a *basilisk*-managed conda environment[40] linked to *hicstuff*, a multipurpose lightweight Hi-C processing Python library[41]. This environment allows *HiCool* to align paired-end sequencing data to a genome reference, parse them into a standard '.pairs' file, filter out invalid pairs[42] and PCR duplicates, bin them into multi-resolution balanced '.mcool' and '.hic' matrix files and automatically generate an HTML report of the processing (Fig. 2a, b). The implementation of *HiCool* as a Bioconductor package enables its efficient integration with other local Hi-C analysis packages (Fig. S1), and unlocks access to genomic databases, e.g. to automatically retrieve and cache cloud-hosted pre-built genome reference indexes by using a genome ID string (e.g. "mm10", "GRZc10", …), accelerating local Hi-C data pre-processing and direct import in R.

## Hi-C visualization

The *HiCExperiment* object inherits methods from the core *GInteractions* and *GRanges* classes to provide a flexible representation of Hi-C data in R. The *HiContacts* package leverages these inheritances to explore *HiCExperiment* objects, focusing on four main topics: Hi-C visualization, contact matrix-centric analysis, interactions-centric analysis and structural feature annotation (Fig. 2A, list of functions in Table S1, interoperability illustrated in Fig. S1).

Hi-C exploratory data analysis is instrumental in generating hypotheses, discovering patterns, and directly answering biological questions. A generic *plotMatrix* function is provided in the *HiContacts* package and can operate on *HiCExperiment*, *GInteractions* or standard matrix objects, with extensive Hi-C-related customization options (Fig. S2). These include single matrix visualization (Fig. S2A, B), side-by-side comparison of two matrices (Fig. S2C), visualization of ratio, observed vs. expected (O/E) and correlation matrices (Fig. S2E–G), support for horizontal Hi-C maps (Fig. S2G), annotation of structural features and alignment with genomic tracks (Fig. S2F), and visualization of aggregated matrices (Fig. S2H). All visualization functions provided by *HiContacts* return *ggplot* objects that can be easily customized, e.g. to change the scaling, range or hue of the color map or to add additional details or labels and generate publication-ready figures.

## Contact matrix-centric analysis

In a Hi-C analysis workflow, a preliminary requirement is the normalization of contact matrices. A well-established approach for matrix normalization is the matrix balancing approach[42,43]. By default, *HiCool* processing performs such normalization automatically, but the end user may need to manually normalize existing contact matrices. The *HiContacts* package implements the balancing of a *HiCExperiment* object, calculating weight scores for each bin and adding a new normalized score metric to each genomic interaction (Fig. 2a).

Several basic matrix operations can be applied to *HiCExperiment* objects. *HiContacts* defines operators to subset or merge Hi-C maps, or to subtract, divide or sum two Hi-C maps. *HiContacts* also provides a random subsampling method of Hi-C interactions that preserves sample-specific distance-dependent interaction frequencies.

Other calculations can be performed on *HiCExperiment* instances. *HiContacts* can estimate the overall expected signal for an imported contact matrix and compute the ratio of observed vs. expected (O/E) interaction frequency (Fig. S2F). *HiContacts* can also compute correlation matrices, to reveal a stereotypical plaid pattern in which interactions are enriched between chromosome segments belonging to the same compartment (AA or BB)[18] (Fig. S2G).

Finally, an operation frequently performed when analyzing Hi-C matrices is the aggregation of matrix snippets, e.g. matrix subsets centered at all topological domain boundaries or all chromatin loops[44]. The *AggrHiCExperiment* class stores and averages the Hi-C signal across a set of snippets of interest. Because *AggrHiCExperiment* is an extension of the core *HiCExperiment* class, it inherits all the methods available to *HiCExperiment* instances, including visualization functionalities (Fig. S2H).

## Interactions-centric analysis

The proportion of cis (intra-chromosomal) and trans (inter-chromosomal) interactions per chromosome can be calculated with *HiContacts* to investigate the propensity of chromosomes to form chromosomal territories with limited intermixing[45] (Fig. S2I).

The chromosome-wide distance-dependent interaction frequency (a.k.a. P(s)) and its slope are valuable metrics that can be used to infer physical properties of individual chromosomes[18,46]. For example, in yeast entering mitosis, there is a significant decrease in interactions in the 20–30 kb range and a downward shift in the P(s) slope beyond this range (Fig. S2J). These features have been successfully used to accurately model the reorganization of the nuclear genomic content into mitotic condensed chromosomes[47–49]. Distance-dependent interaction frequency can also be summarized in scalograms (Fig. S2K): the median interaction genomic distance (±25%, or other quantiles specified by the end user) can be plotted along a linear axis representing a chromosome segment. This is often useful for deciphering the behavior of chromatin interactions along chromosomes[50].

Finally, on a smaller scale, one may also be interested in studying interactions between an discrete viewpoint (a.k.a. bait) locus (e.g. a single or cluster of regulatory elements) and neighboring genomic features (e.g. other regulatory elements, gene bodies, repeats, etc). Profiles of contacts between such a viewpoint and the rest of the genome, sometimes referred to as virtual 4 C plots, can be computed with *HiContacts* (Fig. S2L). This feature is an efficient way to summarize and compare interactions between a number of different loci at once.

## Structural feature annotations

A key step in using chromosome conformation capture data to explore the functional organization of chromosomes is the annotation of structural features. *HiContacts* implements methods to identify A/B chromosome compartments using eigenvector decomposition[18], topologically associated domains (TADs[51]), using a diamond insulation score[52] et chromatin loops using computer vision[53] (Table S1). It is nonetheless advised to investigate structural features using a range of different methods. For instance, A/B compartments can be identified in R with HiTC and HiCDOC packages, while finer sub-compartments can be annotated using CALDER[54–56]. To allow end-users to use best-suited existing R packages, *HiCExperiment* objects can be coerced into the specific data structures such as matrices, data frames or *GInteractions*.

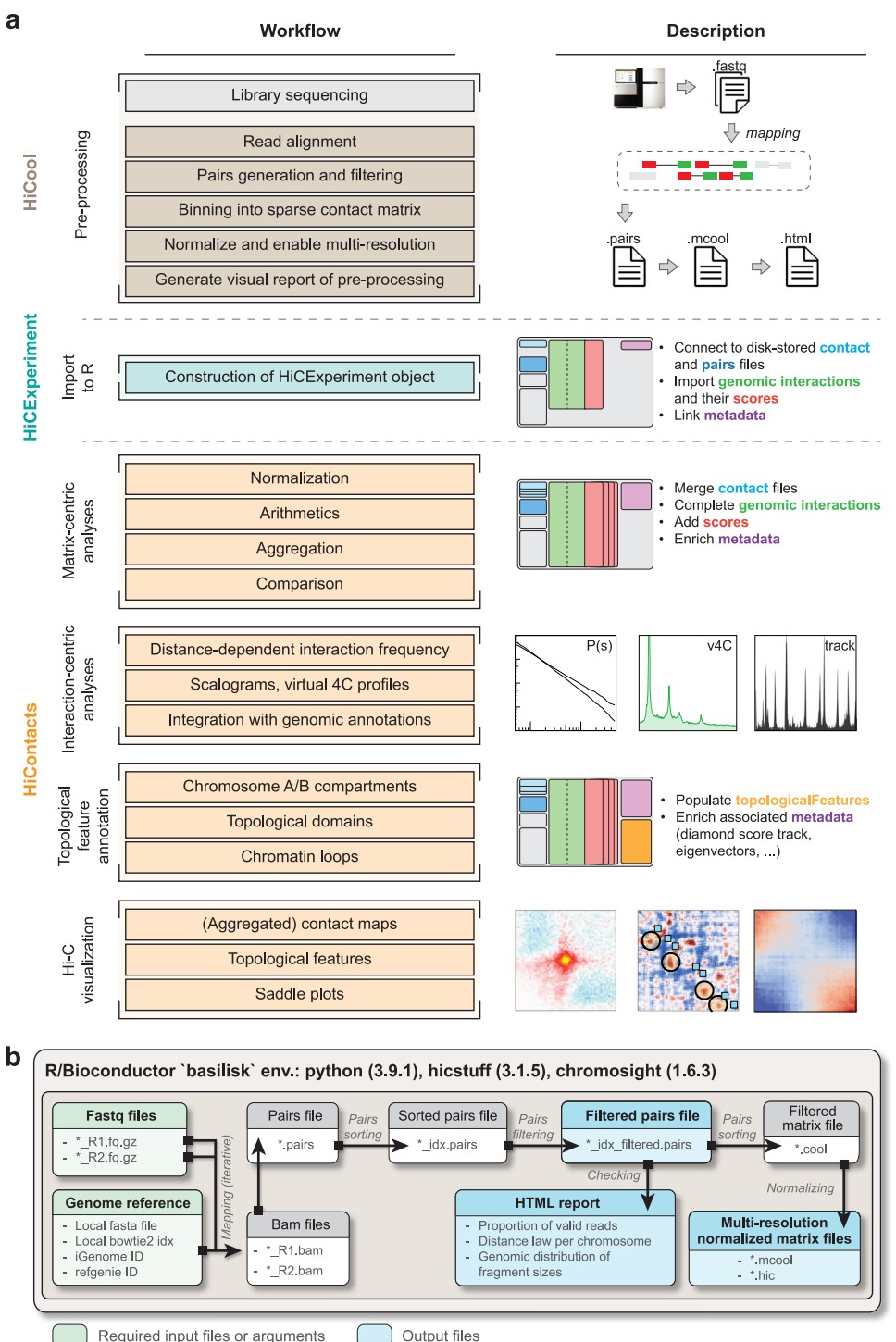

**Fig. 2 | Bioconductor workflow to import and analyze Hi-C data in R. a** Raw Hi-C.fastq files can be pre-processed in R with *HiCool* to generate pairs and binned matrix files. Note that other standard workflows exist to pre-process Hi-C.fastq files in a command-line interface. Binned contact matrices can be imported in R as *HiCExperiment* objects using the import method defined in the *HiCExperiment* package. From there, Hi-C data can be directly visualized, or analyzed from two different perspectives, to perform matrix-centric analyses (e.g. matrix balancing, aggregation, comparison, etc.) or interaction-centric analysis (e.g. cis/trans contact ratios, distance-dependent interaction frequency, interaction profiles, etc.). **b** The *HiCool* package enables Hi-C.fastq processing in R by automatically setting up and temporarily activating a self-managed conda environment linking to *hicstuff*, a multipurpose lightweight Hi-C processing Python library. It internally generates the pairs and binned matrix files, saves a visual summary of the dataset and the processing workflow, and outputs a *ContactFile* and a *PairsFile* objects to facilitate the import of the pre-processed data in R.

## Data integration

Hi-C has gained traction in several fields related to genome biology, and several consortia have developed large-scale programs based on this technique. The NIH 4D Nucleome Program hosts a data portal that lists > 500 chromosome conformation capture experiment sets performed in humans and a variety of model organisms[15]. For each experiment set, Hi-C contact matrices, pairs files, coverage tracks and downstream analysis files are publicly available. The DNA Zoo consortium is using Hi-C scaffolding to generate genome references for hundreds of animals, plants, fungi and microorganisms. The polished

genome sequences and corresponding Hi-C contact maps are directly accessible on a dedicated website[8]. Two R packages, *fourDNData* and *DNAZooData*, provide gateways to these databases. A list of available experiment sets and their metadata is provided within each package, and the actual data files ('.*mcool*', '.*hic*', '.*bw*', '.*bed*', '.*pairs*' and '.*fasta*' files) associated with an experiment of interest can be seamlessly downloaded by providing a sample ID, and are locally cached for reuse across independent R sessions. For example, the mouse and chicken Hi-C data presented in Fig. S2 and Fig. 4 were retrieved directly in R using the *fourDNData* package. Providing programmatic access to existing databases will accelerate investigation in genome biology and open new avenues of research.

## Interoperability between Hi-C packages

The *HiCExperiment* class provides interoperability between Hi-C packages. To illustrate this point, we present here a typical workflow to analyze seven Hi-C yeast datasets obtained from WT cells or mutants of the cohesin complex, synchronized in either G1 or G2/M, a stage in which chromosomes are compacted into arrays of loops[57]. For each library, Hi-C reads are aligned to the yeast reference genome and binned into contact maps using *HiCool*. Visual inspection of Hi-C maps suggests an increase in contact over longer distances in G2/M vs G1, enhanced in the absence of *wpl1* and *wpl1/eco1* (Fig. 3a), and this is confirmed by the P(s) curves (Fig. 3b). That the replicates are coherent is demonstrated by both the strong overlap of the P(s) curves (Fig. 3b) and by the stratum-adjusted correlation coefficients (SCC) calculated using *HiCRep*[58]. Here, we use the *HiCExperiment* coercion methods to convert the imported Hi-C maps into dense matrices, the input class required by *HiCRep*. SCC scores show that replicates for WT G2/M and *wpl1* are overall correlated, while the two replicates for WT G1 are slightly more divergent (Fig. 3c). The stratum-dependent correlation between G1 and G2/M replicates decreases dramatically at short distance (10–30 kb), corresponding to the range of cohesin-mediated chromatin loops along G2/M chromosomes in yeast (Fig. 3d). In contrast, stratum-dependent correlation with *wpl1* single mutant and *wpl1/eco1* double mutant decreases at mid-range (50–100 kb) and mid-to-long-range (50–200 kb) respectively, consistent with the independent roles of Eco1 and Wpl1 factors in chromatin loop formation. We took advantage of the replicates to perform differential interaction (DI) analysis using the *multiHiCcompare* package[59]. Using *HiCExperiment*, we imported w*pl1* and WT replicates chromosome XI Hi-C data and seamlessly coerced them into the *multiHiCcompare*-specific tabular format. The contact frequency fold-change and adjusted *p*-values computed by *multiHiCcompare* are injected back into the original *HiCExperiment* objects to visually represent these metrics in Hi-C maps (Fig. 3e) or as volcano plots, separating inter-arm and intra-arm interactions over chrXI (Fig. 3f). This analysis highlights that the increase in contact frequencies over longer distances occurs specifically for intra-arm contacts in *wpl1* compared to WT, while contacts spanning the acrocentric chrXI centromere decrease. Interoperability with other packages in R is further illustrated in the following page of the companion online book: https://bioconductor.org/books/devel/OHCA/pages/interoperability.html.

## Delivering new biological insights using Hi-C

To illustrate how *HiCExperiment* can be leveraged to raise new biological hypotheses, we investigated a time-course Hi-C dataset of chicken cells released from a G2 block into mitosis[60]. We used the *fourDNData* gateway package to retrieve the data processed by the 4D consortium, and *HiContacts* to annotate compartments. Hi-C maps of chr3 (Fig. 4a) illustrate the progressive loss of compartment organization following G2 release, resulting in a rod-like polymer organization as early as 10 min after release when cells are in prophase and followed by the emergence of a second broader diagonal

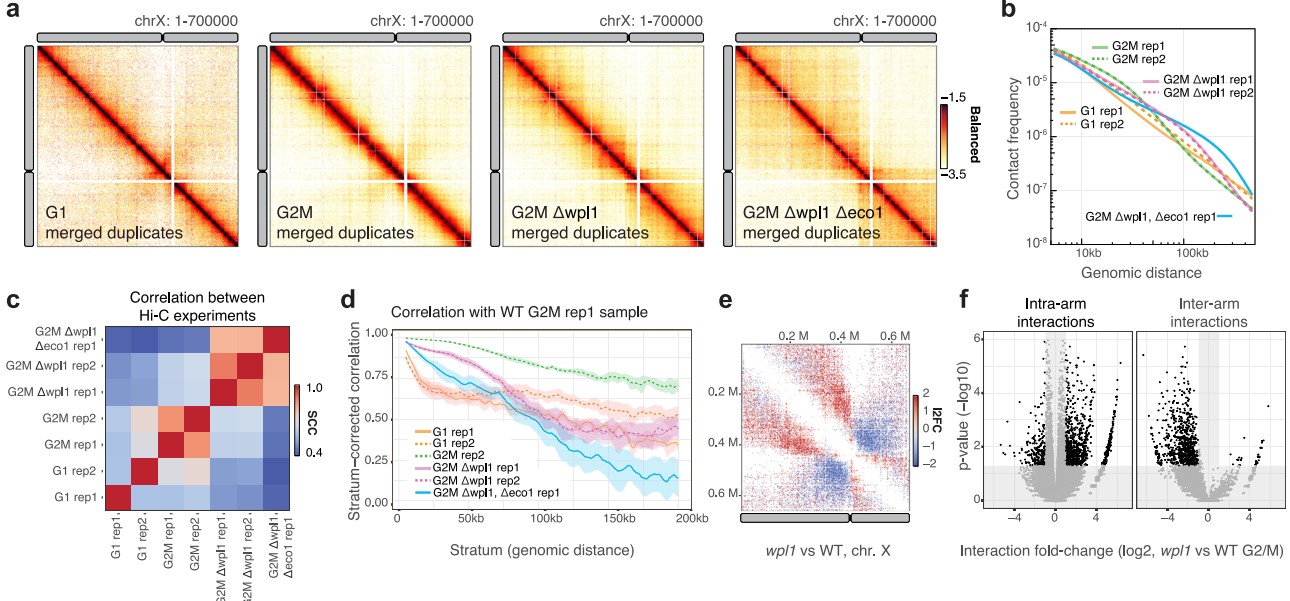

**Fig. 3 | Comparison of distance-dependent interaction frequency across samples. a** Hi-C contact matrix over chromosome X for four different yeast samples: WT G1 merged duplicates, WT G2/M merged duplicates, *wpl1* G2/M merged duplicates and *wpl1/eco1* single replicates (1 kb bins). **b** Distance-dependent contact frequency (P(s)) for each of the seven individual replicates. **c** Overall stratum-corrected correlation (SCC) scores between each of the seven individual replicates, computed with *HiCRep*. The line represents the average SCC across yeast chromosomes. The shaded ribbon represents the 90% confidence interval. **d** Distance-dependent correlation scores between WT G2/M replicate 1 and the other datasets, computed with *HiCRep*. **e** Contact matrix over chromosome X (2 kb bins). The color code represents the interaction frequency fold-change between *wpl1* G2/M and WT G2/M (in log2 scale), computed with *multiHiCcompare*. **f** Volcano plot showing the statistical significance (-log10 *p*-value) versus the magnitude of the interaction frequency fold-change between *wpl1* G2/M and WT G2/M within the chromosome X. *P*-values and fold-change estimates are computed with *multiHiCcompare* using a negative binomial exact test without correction for multiple testing. Intra-arm and inter-arm (centrosome-spanning) interactions are shown in two separate facets. Significant differential interaction (*p*-value ≤ 0.05, absolute fold-change ≥ 2) are highlighted.

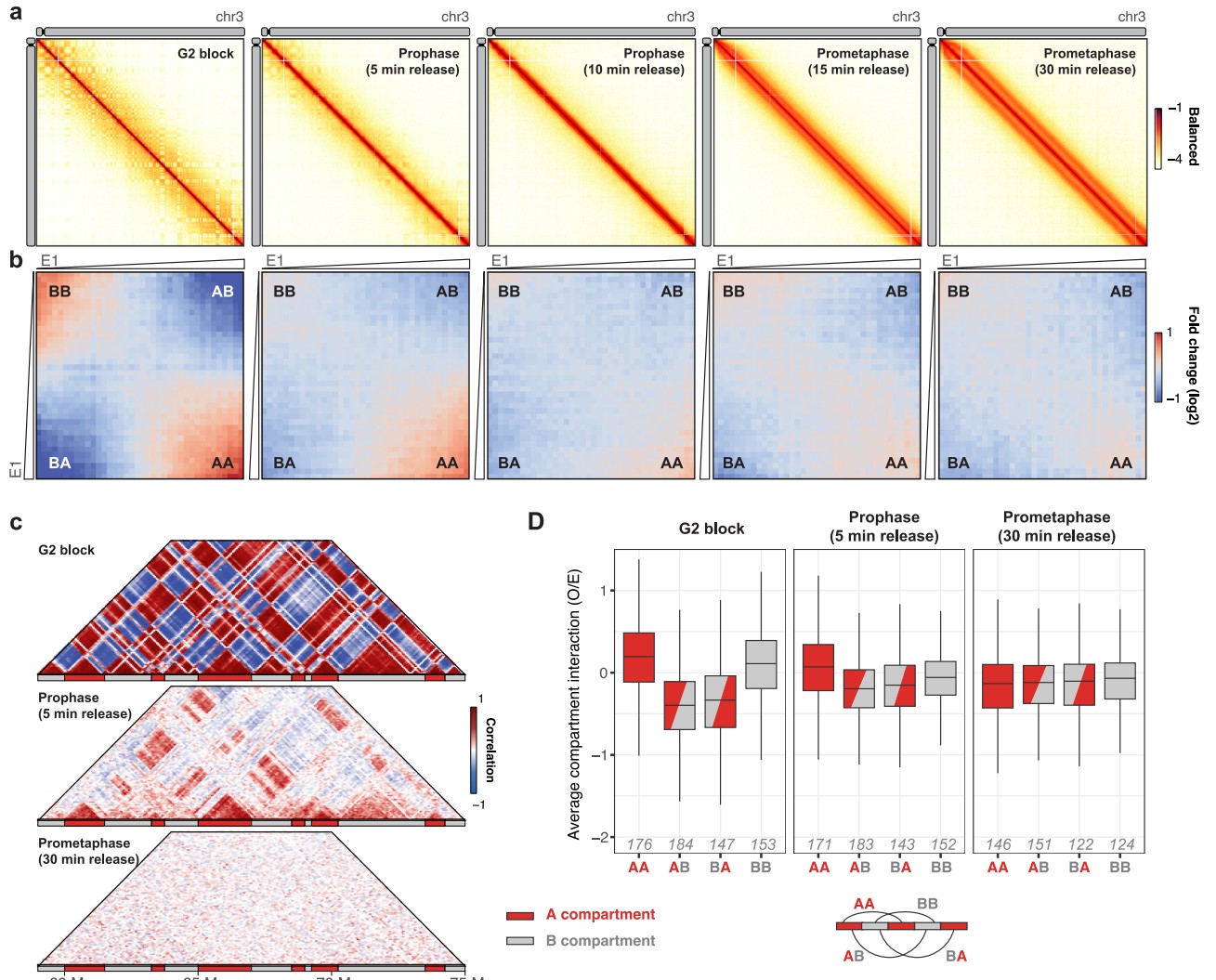

**Fig. 4 | Comparison of A/B interactions during mitotic entry. a** Hi-C contact matrix over chromosome 3 for four different chicken samples: cells blocked in G2, or 5, 10, 15 and 30 min after G2 release (100 kb bins). **b** Saddle plots for the respective Hi-C datasets shown in G. 250 kb-wide non-overlapping genomic windows are grouped in 38 quantiles according to their A/B compartment eigenvector value (E1), from lowest eigenvector values (i.e. strongest B compartments) to highest eigenvector values (i.e. strongest A compartments). The average observed vs. expected (O/E) interaction scores for pairwise eigenvector quantiles are computed and plotted in a 2D heatmap. **c** Hi-C correlation matrices over a 16 Mb-wide segment of chromosome 4, in three different samples: cells blocked in G2, or cells 5 and 30 min after G2 release. The color scale indicates correlation of interaction profiles between any pair of pixels (100 kb bins). The A/B compartments identified in cells blocked in G2 are displayed below each heatmap. Note the loss of positive correlation preferentially between pairs of B compartments 5 min after G2 release. **D** Boxplot of O/E interaction scores between pairs of homologous (A-A or B-B) or heterologous (A-B and B-A) compartments (250 kb bins), in three different samples: cells blocked in G2, or cells 5 and 30 min after G2 release. Only interactions between loci > 5 Mb apart are used. The number of interactions (in thousands) is indicated in italics. The lower and upper hinges correspond to the first and third quartiles (the 25th and 75th percentiles), The lower and upper whiskers extend from the hinge to the smallest/largest value no further than 1.5 * IQR from the hinge (IQR: interquartile range, or distance between the first and third quartiles). Outliers are not displayed here.

corresponding to helical coiling of chromosomes[60]. We generated saddle plots for each time point with *HiContacts* and noted that although AA and BB interactions are comparably lost at 10 min of release onwards, at 5 min BB interactions seem to be specifically depleted compared to AA ones (Fig. 4b). Correlation matrices over a magnified section of chr4 at G2, and 5 min and 30 min after release further confirmed that at this locus AA interactions are retained at 5 min while BB interactions disappear (Fig. 4c). We quantified the average contact frequency between pairs of genomic loci at these three time points, revealing a similar trend genome-wide (Fig. 4D). These results suggest that within minutes after G2 release, the B compartment - corresponding to heterochromatin - is affected faster than A compartment. This is consistent with the model whereby H3 S10 phosphorylation, occurring in late G2 first at chromocenters,

initially induces HP1 eviction from H3K9me3[61] and heterochromatin dissolution[62], and then spreads across entire chromosomes to allow for mitotic condensation[63].

## Discussion

Over the past decade, dozens of Bioconductor-hosted packages have led to widely adopted functional data classes for the generation, parsing and analysis of emerging genomic technologies. These developments allow advanced multi-omics analyses in R to an extent unmatched in other programming languages. Yet, manipulating chromosome conformation capture standard file formats in R remains particularly cumbersome. Here, we present the implementation of a flexible *HiCExperiment* class built on the robust Bioconductor core infrastructure. The *HiCExperiment* class facilitates

chromosome conformation capture data integration (from Hi-C, micro-C, …) into existing genomic analysis workflows in R, reducing redundancy and improving interoperability. The companion *HiContacts* package provides the essential toolkit to compare, aggregate and further investigate *HiCExperiment* objects. A detailed introduction and extensive examples of Hi-C data analysis workflows are provided in the companion website https://bioconductor.org/books/OHCA/.

This rich ecosystem has several advantages compared to existing chromosome conformation capture libraries: (1) it is embedded in the genomics-focus Bioconductor ecosystem, ensuring a rational genomic representation of C data and evolvability (an extension of *HiCExperiment* to support single-cell Hi-C data is currently in development); (2) it extends pre-existing generic methods used by a large community, facilitating the intuitive integration of the C data in existing genomics workflows; (3) it supports quantitative and qualitative analysis of C data, represented as a numerical matrix or as a set of genomic interactions; (4) the daily building/testing infrastructure maintained by Bioconductor assures reproducibility of chromosome conformation capture analyses. For developers, a Docker image with preinstalled development versions of *HiCExperiment*-related packages is available here: https://github.com/users/js2264/packages/container/package/ohca.

A tight integration of *HiCExperiment* within the Bioconductor ecosystem unlocks future development opportunities for Hi-C data analysis. First, *HiCExperiment* could adopt the "tidy" grammar recently adapted to omics data investigation[64], a project spearheaded by the Bioconductor community. This would make Hi-C data wrangling more intuitive and accessible to new investigators. Secondly, Bioconductor supports a *DelayedMatrix* framework and a block processing mechanism, which could be used to improve summarization of Hi-C data over multiple loci. Finally, Bioconductor is currently making efforts to deploy HiCExperiment functionalities within the AnVIL cloud computing platform, a project powered by Terra to facilitate collaborative data investigation between biomedical researchers[65]. We hope that this will accelerate the use of Hi-C in biomedical research, e.g. to shed light on genomic rearrangement events often identified through Hi-C[66,67].

## Methods

All analyses were performed using R 4.3.0 with Bioconductor 3.18. Further details of how each analysis was performed can be found in the Code Availability section.

### Reporting summary

Further information on research design is available in the Nature Portfolio Reporting Summary linked to this article.

## Data availability

All data presented in this manuscript have already been published. Yeast Hi-C data come from[57] and fastq files were obtained from the SRA repository (SRA accession numbers: SRR8769554, SRR10687276, SRR8769549, SRR10687281, SRR8769551, SRR10687278, SRR8769555) or directly obtained through *HiContactsData*. Yeast ChIP-seq data come from[68] and processed data were obtained from GEO (GSM6703614). Chicken Hi-C data come from[60] and was directly imported from the 4DN data portal with *fourDNData* (ExperimentSet accession numbers: 4DNES9LEZXN7, 4DNESNWWIFZU, 4DNESGDXKM2I, 4DNESIR416OW, 4DNESS8PTK6F). micro-C data generated from HFFc6 cells[69] was also imported from the 4DN data portal (ExperimentSet accession number: 4DNESWST3UBH).

## Code availability

All the analysis steps are extensively described as dedicated workflows in the companion website: https://bioconductor.org/books/OHCA/.

Additional examples are also available from the following documentation webpages: Importing Hi-C data (https://js2264.github.io/HiCExperiment/reference/HiCExperiment.html#ref-examples), Arithmetic with Hi-C data (https://js2264.github.io/HiContacts/reference/arithmetics.html#examples) and Plotting Hi-C matrices (https://js2264.github.io/HiContacts/reference/plotMatrix.html). *HiCExperiment* is freely available on Bioconductor (https://bioconductor.org/packages/HiCExperiment), and the source code is hosted on a GitHub repository (https://github.com/js2264/HiCExperiment). *HiContacts*, *HiCool*, *fourDNData* and *DNAZooData* packages are also freely provided as Bioconductor packages (https://bioconductor.org/packages) and publicly hosted on GitHub. *hicstuff* is publicly available as a standalone python package from bioconda.

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

## Acknowledgements
We thank all our colleagues from the laboratory Régulation spatiale des génomes for fruitful discussions. This research was supported by fundings from the European Research Council under the Horizon 2020 Program grant agreement 771813, the Q-life program and the Agence Nationale pour la Recherche to R.K. (ANR-22-CE12-0013-01; ANR-19-CE13-0027-02). J.S. is recipient of a Postdoctoral ARC fellowship.

## Author contributions
Conceptualization: J.S.; Methodology: J.S., C.M.-D.; Software: J.S., C.M.-D., L.B., A.B.; Formal analysis: J.S.; Investigation: J.S.; Resources: J.S., C.M.-D.; Writing—Original Draft: J.S.; Writing—other versions: J.S., R.K.; Visualization: J.S.; Supervision: R.K.; Project administration: R.K.; Funding acquisition: R.K.

## Competing interests
The authors declare no competing interests.
