## [Peer Review File · Nature Communications]

Orchestrating chromosome conformation capture analysis with BioconductorREVIEWER COMMENTS

Reviewer #1 (Remarks to the Author):

The manuscript "Orchestrating chromosome conformation capture analysis with Bioconductor" describes Bioconductor packages, data structures, visualization capabilities, and workflow/tutorial-like case scenarios of various analyses of Hi-C data. In essence, it is a detailed vignette of using the HiCExperiment, HiContacts, HiCool, fourDNData, DNAZooData developed by the authors. The manuscript is short but the online version is extensive and detailed.

- The book introduces many external packages and pipelines. While generally comprehensive, such references are incomplete. For example, when describing the general pipelines for Hi-C data processing, HiCExplorer, HiCUP, FanC analysis pipelines are not mentioned. The GENOVA, Mariner R packages with the similar analysis and visualization functionality are not mentioned. It is not required, and perhaps not possible to refer to all of these, but it may be beneficial to add a reference to more general lists of Hi-C software, such as https://github.com/mdozmorov/HiC_tools, which is also incomplete but covers a broad spectrum of Hi-C tools.

The HiCExperiment, HiContacts, HiCool show build error on Bioconductor.

- The rendering of some references is inconsistent. E.g., "J. O. et al. (2017)", while the more readable one would be "Davies J.O. et al. (2027)". Most references also lack initials, e.g., "Durand et al. (2016)". Also, each page has an empty "References" header at the bottom, but no references until the last page. It is a minor annoyance, OK if impossible to correct in Quarto.

- In the sentence "More information about the conventions related to this text file are provided by the 4DN consortium", suggesting moving the link from "4DN consortium" to "More information" because that's what the link is referring to.

- There are many ideas how this project can be developed. The Discussion is somewhat short and lacks future directions description. Adding them should be beneficial.

Reviewer #2 (Remarks to the Author):

Orchestrating chromosome conformation capture analysis with Bioconductor by J. Serizay and colleagues describe their R code to process and analyse Hi-C data.

The approach developed by the authors combines state of the art approaches for an efficient preprocessing, handling and more high level analysis of chromatin conformation data (limited by R and other software deployed by the functions).

Although the authors do not present novel approaches that would change the approach towards HiC data analysis, I am very much positive about the work presented in this paper.

First of all, it is essential to provide the community with tools that are easily implemented and shared across labs. There are several "pipelines" in the field (the authors take advantage of them). Yet, none of these tools allows to put the analysis so nicely in one framework, easy to implement and (potentially) modify by others. Using R/Bioconductor seems like a perfect way to address this need.

Second, the approaches the authors use seem pretty complete, from preprocessing to the analysis of compartments and domains. The visualisation functions are particularly nice and versatile (saddle plots, virtual 4C).

Third, despite lack of a fundamentally novel approach, the authors introduce a new way of storing HiC data in R which is a great plus as it will allow to intersect datasets more easily. Likewise, I like the infrastructure to query the data available from the 4DN.

Having said all that, it would be important to generate more information that will be useful for the users. The most recent approaches in studying chromatin conformation introduce a dramatic increase in Hi-C resolution. The authors should address the hardware requirements to manipulate data at the resolution of single nucleosomes.

It would perhaps be also interesting to implement sub-compartment calling.

Reviewer #3 (Remarks to the Author):

In this work, Serizay and colleagues describe a Bioconductor-based workflow for end-to-end analysis of chromatin conformation capture (mostly Hi-C) data. They implement data structures to represent Hi-C datasets and genomic contacts on file, as well as methods to perform common manipulations, calculations and visualization. The manuscript is concise and well-written, and the accompanying book is comprehensive and user-friendly. This is likely to be a great resource for the Bioconductor community and I don't have any major concerns about it. Nonetheless, I will throw in some comments so as to prove that I did do my job as a reviewer.

MINOR:

- Be careful about the name of the "HiCExperiment" class. This suggests that it is a SummarizedExperiment subclass (e.g., SingleCellExperiment, SpatialExperiment)... which it isn't. Users may think that it is possible to use SummarizedExperiment methods like assay() and colData(), and be surprised when those fail. Perhaps it may be too late to change it at this point, but I will just say this can become a source of regret later on; for example, the naming of the "InteractionSet" is quite unfortunate, because it is actually an SummarizedExperiment subclass but the naming suggests otherwise.
- Perhaps the authors could explore the use of DelayedArrays to represent contact matrices directly from the various Hi-C file formats, especially those based on HDF5. This would provide a matrix-like API that extracts data on demand, e.g., users could intuitively subset with the usual "[" operators before pulling out the subset of data that they're interested in.
- I am sure that this is already being considered, but it would be worthwhile deploying the book as part of the Bioconductor book releases, e.g., <https://bioconductor.org/books/release/>. This checks that the book runs correctly with the latest versions of all packages, and the Bioconductor domain name is also a bit more official than a random-looking GH pages site.

VERY MINOR:

- A couple of references to basilisk, but this is missing a actual citation AFAICT.
- The book has lots of notes and reminders in alert boxes. I find these mildly distracting at their current frequency.
- diffHic is another (maybe the oldest?) package for differential Hi-C analyses, that you could mention in Chapter 9. As long as you can transform your data to an InteractionSet, it should be good to go.

We would like to sincerely thank the three reviewers for their review of the manuscript. We have made all efforts to systematically address each comment by updating the manuscript and source code accordingly. Here we provide a point-by-point response to each reviewer.

Reviewer #1 (Remarks to the Author):

The manuscript "Orchestrating chromosome conformation capture analysis with Bioconductor" describes Bioconductor packages, data structures, visualization capabilities, and workflow/tutorial-like case scenarios of various analyses of Hi-C data. In essence, it is a detailed vignette of using the HiCExperiment, HiContacts, HiCool, fourDNData, DNAZooData developed by the authors. The manuscript is short but the online version is extensive and detailed.

We thank Reviewer #1 for their careful reading of our manuscript and our supporting online book.

- The book introduces many external packages and pipelines. While generally comprehensive, such references are incomplete. For example, when describing the general pipelines for Hi-C data processing, HiCExplorer, HiCUP, FanC analysis pipelines are not mentioned. The GENOVA, Mariner R packages with the similar analysis and visualization functionality are not mentioned. It is not required, and perhaps not possible to refer to all of these, but it may be beneficial to add a reference to more general lists of Hi-C software, such as https://github.com/mdozmorov/HiC_tools, which is also incomplete but covers a broad spectrum of Hi-C tools.

We thank the reviewer for their suggestion of additional softwares/tools to include in our manuscript. We added a sentence in the introduction of the manuscript (2nd §) to mention HiCExplorer, HiCUP and GENOVA. FAN-C was already mentioned in the introduction. We also added a section referring to the extended list of Hi-C related software from https://github.com/mdozmorov/HiC_tools in the online book.

The HiCExperiment, HiContacts, HiCool show build error on Bioconductor.

This issue has been reported to the Bioconductor team and is currently being investigated. That being said, this does not prevent the end-user from installing the packages, as the ERROR raised by the Bioconductor Build System are independent from the functions defined in these packages.

- The rendering of some references is inconsistent. E.g., "J. O. et al. (2017)", while the more readable one would be "Davies J.O. et al. (2027)". Most references also lack initials, e.g., "Durand et al. (2016)". Also, each page has an empty "References" header at the bottom, but no references until the last page. It is a minor annoyance, OK if impossible to correct in Quarto.

We thank the reviewer for reporting these inconsistencies in the online book. We have now made our best efforts to correct them throughout the manuscript and the online book.

- In the sentence "More information about the conventions related to this text file are provided by the 4DN consortium", suggesting moving the link from "4DN consortium" to

"More information" because that's what the link is referring to.

This has been fixed in the online book.

- There are many ideas how this project can be developed. The Discussion is somewhat short and lacks future directions description. Adding them should be beneficial.

We have added the following paragraph to the Discussion:

“A tight integration of *HiCExperiment* within the Bioconductor ecosystem unlocks future development opportunities for Hi-C data analysis. First, *HiCExperiment* could adopt the “tidy” grammar recently adapted to omics data investigation (Hutchison et al. 2023), a project spearheaded by the Bioconductor community. This would make Hi-C data wrangling more intuitive and accessible to new investigators. Secondly, Bioconductor supports a *DelayedMatrix* framework and a block processing mechanism, which could be used to improve summarization of Hi-C data over multiple loci. Finally, Bioconductor is currently making efforts to deploy *HiCExperiment* functionalities within the AnVIL cloud computing platform, a project powered by Terra to facilitate collaborative data investigation between biomedical researchers (Schatz et al. 2022). We hope that this will accelerate the use of Hi-C in biomedical research, e.g. to shed light on genomic rearrangement events often identified through Hi-C (Lupiáñez et al. 2015; Melo et al. 2020).”

Reviewer #2 (Remarks to the Author):

Orchestrating chromosome conformation capture analysis with Bioconductor by J. Serizay and colleagues describe their R code to process and analyse Hi-C data.

The approach developed by the authors combines state of the art approaches for an efficient preprocessing, handling and more high level analysis of chromatin conformation data (limited by R and other software deployed by the functions).

Although the authors do not present novel approaches that would change the approach towards HiC data analysis, I am very much positive about the work presented in this paper.

First of all, it is essential to provide the community with tools that are easily implemented and shared across labs. There are several "pipelines" in the field (the authors take advantage of them). Yet, none of these tools allows to put the analysis so nicely in one framework, easy to implement and (potentially) modify by others. Using R/Bioconductor seems like a perfect way to address this need.

Second, the approaches the authors use seem pretty complete, from preprocessing to the analysis of compartments and domains. The visualisation functions are particularly nice and versatile (saddle plots, virtual 4C).

Third, despite lack of a fundamentally novel approach, the authors introduce a new way of storing HiC data in R which is a great plus as it will allow to intersect datasets more easily. Likewise, I like the infrastructure to query the data available from the 4DN.

We thank Reviewer #2 for their appreciation of our work and their helpful comments.

Having said all that, it would be important to generate more information that will be useful for the users. The most recent approaches in studying chromatin conformation introduce a dramatic increase in Hi-C resolution. The authors should address the hardware requirements to manipulate data at the resolution of single nucleosomes.

This is a point we did not address in the original manuscript, and following the reviewer suggestions, we added a few sentences in the "Data representation" section of our revised manuscript to discuss the hardware required to manipulate chromosome conformation capture data at increasing resolutions.

"Thanks to ever-decreasing sequencing costs and improving technology, the average size of chromosome conformation capture datasets is continuously increasing, both in sequencing depth and in resolution. HDF5-derived `.(m)cool` and binary `.hic` files both efficiently store such large-scale data, and `HiCExperiment` objects instantiated from these file formats benefit from efficient parsing libraries based on C code, optimized for speed. Furthermore, because random access is supported for these file formats, contact matrices can be partially imported in R, allowing manipulation of large datasets – such as deeply sequenced micro-C datasets – even on personal laptops with standard hardware configuration (e.g. 4 CPUs and 8-16Gb RAM)."

It would perhaps be also interesting to implement sub-compartment calling.

Several libraries (available in R or from other programming languages) already provide these specific functionalities and are already widely used. For this reason, we believe that (re)implementing these functionalities in yet another library would not positively contribute to improving Hi-C analysis overall. Instead, we added a sentence to mention existing libraries dedicated to sub-compartment calling in R.

“It is nonetheless advised to investigate structural features using a range of different methods. For instance, A/B compartments can be identified in R with HiTC and HiCDOC packages, while finer sub-compartments can be annotated using CALDER. To allow end-users to use best-suited existing R packages, *HiCExperiment* objects can be coerced into the specific data structures such as matrices, data frames or *GInteractions*.”

Reviewer #3 (Remarks to the Author):

In this work, Serizay and colleagues describe a Bioconductor-based workflow for end-to-end analysis of chromatin conformation capture (mostly Hi-C) data. They implement data structures to represent Hi-C datasets and genomic contacts on file, as well as methods to perform common manipulations, calculations and visualization. The manuscript is concise and well-written, and the accompanying book is comprehensive and user-friendly. This is likely to be a great resource for the Bioconductor community and I don't have any major concerns about it. Nonetheless, I will throw in some comments so as to prove that I did do my job as a reviewer.

We would like to thank Reviewer #3 for their comments and their suggestions to improve the manuscript, code base and online documentation.

MINOR:

- Be careful about the name of the "HiCExperiment" class. This suggests that it is a SummarizedExperiment subclass (e.g., SingleCellExperiment, SpatialExperiment)... which it isn't. Users may think that it is possible to use SummarizedExperiment methods like assay() and colData(), and be surprised when those fail. Perhaps it may be too late to change it at this point, but I will just say this can become a source of regret later on; for example, the naming of the "InteractionSet" is quite unfortunate, because it is actually an SummarizedExperiment subclass but the naming suggests otherwise.

We thank the reviewer for raising the fact that the name *HiCExperiment* could be misleading, due to its semantic resemblance with other **Experiment* objects. Unfortunately, the package has already been through the Bioconductor review system and is now installed by 500+ individual end users, and changing the package name and main class appears to be impossible. We hope the clear documentation for this class, available both in vignettes and on our online book, and which notably focuses on supported getters and setters, will be enough to avoid confusion.

- Perhaps the authors could explore the use of DelayedArrays to represent contact matrices directly from the various Hi-C file formats, especially those based on HDF5. This would provide a matrix-like API that extracts data on demand, e.g., users could intuitively subset with the usual "[" operators before pulling out the subset of data that they're interested in.

Delayed parsing of on-disk files is indeed something we would like to implement in the future. Unfortunately, the 2 main file formats for Hi-C data (cool and hic) are not standard array-like file formats (even though the cool format is based on HDF5). As we are not the authors/developers of these file formats, it is beyond our scope to ensure that these file formats evolve towards an architecture compatible with a DelayedArray backend API. This limitation in the two file formats was one of the motivations that led us to implement two separate subsetting methods (subset while parsing with "focus = ..." and subset after parsing with usual "["). If/when these two file formats evolve towards compatibility with the DelayedArray infrastructure, we will add a DelayedArray backend to *HiCExperiment* to unify the subsetting approach for the end-user.

- I am sure that this is already being considered, but it would be worthwhile deploying the book as part of the Bioconductor book releases, e.g., <https://bioconductor.org/books/release/>. This checks that the book runs correctly with the latest versions of all packages, and the Bioconductor domain name is also a bit more official than a random-looking GH pages site.

Migrating the source code of the online book to Bioconductor is currently a work in progress, but will take some time to be implemented. Automatic redirection will be implemented to ensure that the updated URL is available to readers.

VERY MINOR:

- A couple of references to basilisk, but this is missing a actual citation AFAICT.

We have added a citation for the basilisk publication to the manuscript. We thank the reviewer for pointing out this oversight.

- The book has lots of notes and reminders in alert boxes. I find these mildly distracting at their current frequency.

Following discussions with several end-users from different backgrounds, with little to moderate experience with Hi-C and Bioconductor, it appears that such alert boxes are much appreciated, as they provide helpful messages and tips. However, we agree with the reviewer that too many of these boxes were present in the online book. When appropriate, we have moved the content of alert boxes into the body of the web pages rather than in distracting, interruptive boxes.

- `diffHiC` is another (maybe the oldest?) package for differential Hi-C analyses, that you could mention in Chapter 9. As long as you can transform your data to an `InteractionSet`, it should be good to go.

We thank the reviewer for this great suggestion. We added a section in Chapter 9 to demonstrate how `HiCExperiment` objects could be coerced into `InteractionSet` and used in `diffHiC` workflow (see <https://js2264.github.io/OHCA/interoperability.html>) .

REVIEWERS' COMMENTS

Reviewer #1 (Remarks to the Author):

All comments have been addressed.

Reviewer #2 (Remarks to the Author):

I would like to thank the authors for addressing my comments. I feel that my requests and doubts were addressed.

Reviewer #3 (Remarks to the Author):

The authors have addressed all of my concerns and I don't have much to add here.